# Comprehensive Analysis of *NAC* Genes Reveals Differential Expression Patterns in Response to *Pst* DC3000 and Their Overlapping Expression Pattern during PTI and ETI in Tomato

**DOI:** 10.3390/genes13112015

**Published:** 2022-11-02

**Authors:** Songzhi Xu, Zhiyao Zhang, Jiajing Zhou, Xiao Han, Kun Song, Haiying Gu, Suqin Zhu, Lijun Sun

**Affiliations:** 1School of Life Sciences, Nantong University, Nantong 226019, China; 2School of Public Health, Institute of Analytical Chemistry for Life Science, Nantong University, Nantong 226019, China; 3Key Lab of Landscape Plant Genetics and Breeding, Nantong 226019, China

**Keywords:** tomato, SlNAC transcription factors, *Pst* DC3000, phytohormone treatment, overlapping expression pattern

## Abstract

NAC (NAM/ATAF/CUC) transcription factors belong to a unique gene family in plants, which play vital roles in regulating diverse biological processes, including growth, development, senescence, and in response to biotic and abiotic stresses. Tomato (*Solanum lycopersicum*), as the most highly valued vegetable and fruit crop worldwide, is constantly attacked by *Pseudomonas syringae* pv. *tomato* DC3000 (*Pst* DC3000), causing huge losses in production. Thus, it is essential to conduct a comprehensive identification of the *SlNAC* genes involved in response to *Pst* DC3000 in tomato. In this study, a complete overview of this gene family in tomato is presented, including genome localization, protein domain architectures, physical and chemical features, and nuclear location score. Phylogenetic analysis identified 20 *SlNAC* genes as putative stress-responsive genes, named *SSlNAC 1*–*20*. Expression profiles analysis revealed that 18 of these 20 *SSlNAC* genes were significantly induced in defense response to *Pst* DC3000 stress. Furthermore, the RNA-seq data were mined and analyzed, and the results revealed the expression pattern of the 20 *SSlNAC* genes in response to *Pst* DC3000 during the PTI and ETI. Among them, *SSlNAC3*, *SSlNAC4*, *SSlNAC7*, *SSlNAC8*, *SSlNAC12*, *SSlNAC17*, and *SSlNAC19* were up-regulated against *Pst* DC3000 during PTI and ETI, which suggested that these genes may participate in both the PTI and ETI pathway during the interaction between tomato and *Pst* DC3000. In addition, *SSlNAC* genes induced by exogenous hormones, including indole-3-acetic acid (IAA), abscisic acid (ABA), salicylic acid (SA), and methyl jasmonic acid (MeJA), were also recovered. These results implied that *SSlNAC* genes may participate in the *Pst* DC3000 stress response by multiple regulatory pathways of the phytohormones. In all, this study provides important clues for further functional analysis and of the regulatory mechanism of *SSlNAC* genes under *Pst* DC3000 stress.

## 1. Introduction

In nature, plants are often attacked by pathogens in the process of growth and development [1]. Usually, phytopathogens may reduce the biomass, or even kill the plant, which will ultimately affect plant growth and crop yield [1,2]. Over the long term, plants evolved a series of defense mechanisms to adapt and resist various pathogens attack, including physical barriers, antimicrobial compounds [3], and the plant immune system, including pattern-triggered immunity (PTI), as the first tier of the plant immune system, and effector-triggered immunity (ETI). Previous studies demonstrated that reactive oxygen species (ROS) signaling, phytohormone signaling, changes in redox status, inorganic ion fluxes, and transcriptional reprogramming, etc., were triggered during PTI and ETI [4,5]. Among these complex molecular regulatory networks, transcription factors act important roles by binding to promoters of resistance genes and regulating multiple specific stress responsive genes.

NAC (NAM, ATAF, and CUC) transcription factors (TFs), as one of the largest family of transcription factors in plants, comprise a complex plant-specific superfamily and presents in a wide range of species [4,6]. NAC TFs were found to function in various growth and development processes, including leaf or flower senescence [7,8,9], formation of cell wall [10,11], fruit ripening [12,13,14], root growth [15], etc. Besides, a large number of studies demonstrated that NAC TFs played important roles in abiotic and biotic stress responses, especially in crops. For instance, in rice, transcriptional profiling analysis revealed that a total of 63 *ONAC* genes exhibited overlapping expression patterns under various abiotic and biotic stresses [16]. Over-expression of stress-responsive *OsNAC2* can enhance drought or salinity tolerance [17], while knockout of *OsNAC006* resulted in drought and heat sensitivity [18]. Over-expression of *OsNAC58* [19] and *OsNAC60* [20] increased resistance to pathogen attack. In wheat, the most TF members, 559, have been reported [21]. *TaNAC1* [22], *TaNAC2* [23], *TaNAC6s* [24], *TaNAC30* [25], and *TaNACL-D1* [26] were identified as negative or positive regulators in resisting biotic stress by over-expression or knockdown approaches, while over-expression of *TaNAC48* and *TaNAC29* significantly enhanced drought and salt tolerance, respectively [27,28]. Besides, Xia et al. (2010) proved that *TaNAC8* can be significantly induced by biotic stress, abiotic stresses, MeJA (methyl jasmonic acid), and ET (ethylene) [29]. In soybean (*Glycine max*), over-expression of *GmNAC019* and *GmNAC06* significantly improved drought and salt tolerance, respectively [30,31], while *GmNAC42-1* [32] and *GmDRR1* [33] were reported to be involved in plant disease resistance by over-expression or the CRISPR-Cas9 system.

On the other hand, expression of *NAC* genes can also be induced by phytohormones, and some studies have demonstrated that NAC TFs may influence different developmental programs and respond to abiotic and biotic stresses through influencing synthesis and metabolism of phytohormones. For instance, Liu et al. (2018) reported three ABA (abscisic acid)-responsive elements, two SA (salicylic acid)-responsive elements, and two MeJA-responsive elements in the prompter sequence of *ONAC066* [34]. Huang et al. (2017) detected many cis-elements involved in hormones response, including the SA response element, auxin-responsive element, and ABA-responsive element [35]. Mao et al. (2020) indicated that *OsNAC2* plays an important role in root development as an upstream integrator of auxin and cytokinin signals [36]. Liu et al. (2018) revealed that *ONAC066* positively regulates disease resistance by modulating the ABA signal pathway in rice [34]. Du et al. (2014) demonstrated that JA2L, one of the NAC TFs, is more susceptible to *Pseudomonas syringae* pv. *tomato* DC3000 (*Pst* DC3000) by promoting the metabolic genes of SA, *SAMT1*, and *SAMT2*, which suppressed accumulation of SA and led to reopening of stomata [37]. Zhu et al. (2019) demonstrated that *DRL1*, encoding a NAC transcriptional factor, acts as a negative regulator of leaf senescence by regulating ABA synthesis [38]. All in all, identifying *NAC* genes induced by phytohormones is helpful for illustrating the mechanisms of NAC-mediated stress resistance and interactions between NAC TFs and phytohormones under stress.

Tomato (*Solanum lycopersicum*), as one of the most consumed and economically important vegetable and fruit crop worldwide, has made a huge contribution to the human diet [4,39,40]. Meanwhile, tomato is constantly challenged by varieties of biotic and abiotic stress. To date, several NAC TFs, including SlNAC1 [41], SlNAC3 [42], SlNAC4 [43], SlNAM1 [44,45], SlSRN1 [46], and JUB1 [47,48], were identified as regulators in response to drought, salinity, chilling, or heat stress. Except for abiotic stress, tomato is extremely vulnerable to fungi, viruses, and bacteria. *Pst* DC3000, as a destructive disease pathogen, could cause bacterial leaf spot, resulting in yield loss and reduced quality of tomato fruit [49], but effective control measures are still lacking [50]. To clarify the resistance mechanism in response to *Pst* DC3000 in tomato could provide a new strategy in defending bacterial leaf spot.

Although several *SlNAC* genes, such as *SlSRN1*, *SlNAC1*, and *JA2L*, have been characterized in response to *Pst* DC3000 [37,46,51], the expression patterns, functional surveys, and regulatory mechanisms of most *SlNAC* genes under *Pst* DC3000 stress remain largely unclear. Collectively, the aims of this study are to (1) identify the stress-responsive *SlNAC* (S*SlNAC*) genes by phylogenetic analysis; (2) obtain the expression levels of these S*SlNAC* genes under *Pst* DC3000 inoculation; (3) understand the expression patterns of these *SSlNAC* genes under *Pst* DC3000 inoculation during PTI and ETI; and (4) illustrate the expression levels of these *SSlNAC* genes under exogenous hormones treatment, including IAA (indole-3-acetic acid), ABA, SA, and MeJA). The results of this study will provide clues for further functional analysis and of the regulatory mechanisms of the *SlNAC* genes under *Pst* DC3000 stress, which are essential in developing transgenic tomato with enhanced tolerance to *Pst* DC3000 stress.

## 2. Materials and Methods

### 2.1. Sequence Retrieval and Sequence Analysis of Tomato NAC Transcription Factors

To obtain members of the tomato NAC transcription factor gene family, multiple database searches were performed. The Database of Sol Genomics Network (https://solgenomics.net/, accessed on 1 March 2021) and National Centre for Biotechnology Information (NCBI; http://www.ncbi.nlm.nih.gov/, accessed on 1 March 2021) were used to search for members of the *NAC* gene family. In addition, the chromosome, sites, location, and protein length of the tomato *NAC* genes were obtained. Protein domain architectures of the NAC transcription factors were analyzed in the SMART database (http://smart.embl-heidelberg.de/, accessed on 1 April 2021). The physical and chemical features of the NAC protein, including the molecular weight and isoelectric point, instability index, aliphatic index, and grand average hydropathicity, were obtained from protParam (http://web.expasy.org/protparam/, accessed on 1 April 2021). The subcellular localization was predicted by Protcomp 9.0 (http://linux1.softberry.com/berry.phtml?, accessed on 7 May 2021).

### 2.2. Phylogenetic Analysis, Gene Structure, and Conserved Motif Prection of SlNAC Genes

Multiple sequence alignment of full-length protein sequences of tomato NAC transcription factors was performed using the Clustal W2 program with the default parameters. A phylogenetic tree was plotted using MEGA software version 5.05 (Auckland, New Zealand) by the neighbor-joining method with 1000 bootstrap replicates. The exon/intron structures of the *SlNAC* genes were visualized using TBtools software version 1.098769 (Guangzhou, China). To analyze the functional regions of the SlNAC proteins, conserved motifs in full length tomato NAC proteins were identified using the Multiple Expectation Maximization for Motif Elicitation (MEME) program version 4.6.1 (http://meme.nbcr.net/meme/, accessed on 5 June 2021) with the default parameters.

### 2.3. Chromosomal Location and Collinearity Analysis

To identify the chromosomal positions of 101 tomato *NAC* genes, TBtools software version 1.098769 (Guangzhou, China) was used to locate and visually map the *SlNAC* genes on 13 chromosomes in tomato based on the General Feature Format (GFF) information. To further explore the gene duplication events of 101 *SlNAC* genes, collinearity analysis was performed using MCscanX and the gene duplication events were visualized using Advanced Circos in TBtools software version 1.098769 (Guangzhou, China).

### 2.4. Mining and Analysis of RNA-Seq Data-Based Expression Profiling Data

To understand the performance of *SSlNAC* genes resisting against *Pst* DC3000 during PTI and ETI, RNA-seq data available on the Tomato Functional Genomics Database (http://ted.bti.cornell.edu/cgi-bin/TFGD, accessed on 1 July 2021) was mined and analyzed in this study. Data from accessions D007 (Transcriptome sequencing of tomato leaves treated with different bacteria and PAMPs) [52], D010 (Transcriptome sequencing of leaves of resistant (RG-PtoR) and susceptible (RG-prf3 and RG-prf19) tomato plants treated with *Pst* DC3000) [53], D011 (Transcriptome sequencing of tomato leaves treated with different *Pst* DC3000 mutant strains) [53], and D013 (Domain-wise effect of AvrPto on the tomato transcriptome 6 h post inoculation) [54] was downloaded. Then, data related to *Pst* DC3000 treatments were selected and analyzed for the expression patterns of *SlNAC* genes during PTI and ETI by heat maps.

### 2.5. Plant Growth and Treatments

According to previous studies [46,55,56], the tomato cultivar Suhong 2003 was used in this study. Tomato plants were grown in a growth chamber (24 °C 14 h light/20 °C 10 h dark). For analysis of gene expression in response to *Pst* DC3000, four-week-old tomato seedlings were infiltrated with *Pst* DC3000 suspensions according to a previously reported procedure [46,55,56]. Specifically, *Pst* DC3000 was cultured on King’s B (KB) medium containing 50 mg/L rifampicin overnight until the OD_600 nm_ was 0.6 to 0.8 (OD 0.1 = 10^8^ cfu mL^−1^). Then, *Pst* DC3000 was diluted with 10 mM Mgcl_2_ to concentrations of 10^7^ cfu mL^−1^. Before infiltration, *Pst* DC3000 was re-suspended in 10 mM Mgcl_2_ to OD_600 nm_ = 0.0002. For analysis of gene expression in response to biotic stress, four-week-old tomato seedlings were infiltrated under vacuum with *Pst* DC3000 suspensions. Plants infiltrated with a 10 mM Mgcl_2_ buffer solution were used as mock-inoculated controls. Then, the inoculated and control seedlings were kept in a growth chamber at high humidity. The samples were harvested at 24 °C for 6, 12, and 24 h, respectively. For analysis of gene expression in response to defense signaling hormones, 4-week-old tomato plants were treated by foliar spraying with 1.5 mM SA (Sigma-Aldrich, St. Louis, MO, USA), 0.1 mM MeJA (Sigma-Aldrich, St. Louis, MO, USA), 0.1 mM ABA (Sigma-Aldrich, St. Louis, MO, USA), and 0.1 mM IAA (Sigma-Aldrich, St. Louis, MO, USA) at 24 °C for 3, 6, and 12 h. Four-week-old tomato plants were treated by foliar spraying with water as control. The samples were harvested separately and stored at −80 °C until use.

### 2.6. RT-qPCR Analysis

For quantitative real-time PCR analysis, total RNA was extracted from the frozen tomato leaf samples with RNAiso Plus (Takara, Dalian, China), according to the manufacturer’s instructions. The resulting RNA samples were treated with RNase-free DNase (TaKaRa, Dalian, China) and first-strand cDNAs were synthesized using AMV reverse transcriptase (TaKaRa, Dalian, China), following the manufacturer’s instructions. RT-qPCR reactions were performed as previously described [57]. The β actin gene and 20 primers synthesized for the 20 *SSlNAC* genes are listed in Appendix A. Expression data for each *SSlNAC* gene under different treatments were normalized using the expression data of the actin gene as an internal reference. The relative expression level was calculated by the comparative ΔΔCt method. The ΔCt and ΔΔCt were calculated by the formulas ΔCt (target gene) = Ct (target gene) − Ct (actin) and ΔΔCt = ΔCt (treated sample) − ΔCt (untreated sample). Expression folds for each *SSlNAC* gene under a given stress were calculated by the formulas of fold = 2^−ΔΔCt^. Standard errors of the means from three independent biological replicates were calculated.

### 2.7. Statistical Analysis

The mean value represents the relative expression folds between the treatment and control. The error line represents the standard deviation. Student’s *t*-test was carried out to ascertain any significant difference between the different treatments and control (n = 3; * *p* < 0.05 and ** *p* < 0.05) using SPSS software version 19.0 (New York, NY, USA).

## 3. Results

### 3.1. Characterization of the Physical and Chemical Properties of SlNAC Transcription Factors

Jin et al. (2014) updated the plant Transcription Factor database PlantTFDB to version 3.0 (http://planttfdb.cbi.pku.edu.cn, accessed on 1 March 2021) [58]. By searching the Plant Transcription Factor Database (http://planttfdb.cbi.pku.edu.cn/index.php, accessed on 1 March 2021) and SOL Database (http://solgenomics.net/, accessed on 1 March 2021), a total of 101 NAC TFs were obtained by analyzing the structure of the NAC conservative domain. Information on the sequence of amino acids, the length of the amino acid, length of the cDNA, molecular weight, isoelectric point, instability index, aliphatic index, grand average hydropathicity, and nuclear location score are shown in Appendix A. For further convenience, these *SlNAC1–101* genes were named according to the previously proposed nomenclature system (Appendix A, Gene name I).

### 3.2. The Phylogenetic Tree of the NAC Genes and Conserved Protein Motifs of SlNACs

To predict the function of the NAC proteins of tomato, a phylogenetic tree was constructed using *SlNAC* genes from tomato and identified *NAC* genes in rice and *Arabidopsis*. Results showed that the tomato *SlNAC* genes were divided into 13 subtribes (renamed group 1–group 13), as shown in Figure 1. A total of 20 *SlNAC* genes, from group 3, group 6, and group 9, are in the same branch with stress responsive genes in rice and *Arabidopsis*, such as *SNAC1*, *SNAC2*/*OsNAC6*, *OsNAC5*, *ANAC019*, *ANAC055*, *ONAC131*, *ONAC122*/*OsNAC10*, *NTL9*, etc., [16,57,59,60,61,62,63], which indicates these 20 *SlNAC* genes may be related to stress response (named *SSlNAC1*–*SSlNAC20*). Besides, previous studies have proved that *CUC1*, *CUC2*, and *CUC3* are mainly related to the formation of the leaf primordium and flower organ in *Arabidopsis thaliana*, while the NAC transcription factors, such as NST1, NST2, NST3, VND6, and VND7, are related to secondary wall thickening [64,65]. Therefore, the 23 *SlNAC* genes were in the same clades (group 12 and group 13) as the *VND*, *NST*, and *CUC* genes, which indicated that these genes may play important roles in the growth and development of tomato. In the present study, 20 *SSlNAC* genes from group 3, group 6, and group 9 were selected for the next analysis. These *SlNAC1*–*101* genes were also named according to the phylogenetic tree (see Appendix A, Gene name II).

### 3.3. Gene Structures and Conserved Protein Motifs of the SlNACs

The 101 SlNAC proteins were clustered into 13 clades, which are consistent with the phylogenetic relationships between tomato, rice, and *Arabidopsis* (Figure 1 and Figure 2a). Then, to investigate the functional diversity of the SlNAC proteins, a total of 12 conserved motifs were predicted and named as motifs 1–12 (Figure 2b). The results showed that almost all of NAC proteins have motif 2, motif 5, and motif 6. In addition, the NAC proteins in the same group usually had similar motif compositions. For instance, most SlNACs in subgroup 3, subgroup 12, and subgroup 13 consisted of motif 1–6, except for SlNAC45 and SlNAC63, which only have motif 2 and motif 5. A unique motif (motif 7) was only identified in five SlNAC proteins from subgroup 11. Furthermore, the untranslated regions (UTR), CDs, and introns were identified to explore the diversity of the gene structure (Figure 2c). The results showed that the number of exons in the 101 *SlNAC* genes ranged from 1 to 17. *SlNAC19* contained the largest number of exons. Some *SlNAC* genes contained CDs without UTR and introns, and these genes mainly clustered in the same group, which indicated that the *SlNAC* genes in the same clade shared a similar gene structure.

### 3.4. Chromosomal Distribution and Collinearity Analysis of the SlNAC Genes

The location information of the *SlNAC* genes was obtained based on the genome annotation file from the NCBI database. The results showed that 101 *SlNAC* genes were unevenly distributed on 13 chromosomes and were named *SlNAC1*–*SlNAC101* (Appendix A, Figure 3). Chr02 contained the largest number of *SlNAC* genes (17 *SlNACs*), followed by Chr06 (14 *SlNACs*), while chr0 only contained *SlNAC1*. Furthermore, a positive correlation between the number of *SlNAC* genes and the length of chromosome were not detected. To explore the expansion patterns of *NAC* genes in tomato, intragenomic collinearity analysis was carried out. The results indicated that 17 gene pairs were identified. Chr06 contained the largest number of duplicated gene pairs (6 pairs), while Chr00, Chr01, and Chr09 had no duplicated gene pair.

### 3.5. Expression Patterns of 20 SSlNAC Genes under Pst DC3000 Inoculation

To confirm the expression levels of the 20 *SSlNAC* genes under *Pst* DC3000 stress, RT-qPCR was carried out after inoculation with *Pst* DC3000. Compared with the control, 18 *SSlNAC* genes (except for *SSlNAC7* and *SSlNAC19*) have an obvious response to *Pst* DC3000 in different time points after inoculation (Figure 4). For instance, *SSlNAC6* could be very significantly induced at 6, 12, and 24 h post inoculation (hpi) (** *p* < 0.01). The expression levels of *SSlNAC5* increased significantly at 6 and 12 hpi, while expression levels of *SSlNAC9* and *SSlNAC11* increased most significantly at 12 and 24 hpi. *SSlNAC1*, *SSlNAC8*, and *SSlNAC20* could be significantly induced or suppressed at a different stage after *Pst* DC3000 treatment, which indicated that these genes play different roles during the early and late response stage. Expression of *SSlNAC2* and *SSlNAC3* were significantly suppressed at 6 and 24 hpi, respectively, indicating that these genes may act as negative regulators in response to *Pst* DC3000.

### 3.6. Expression Patterns of SSlNAC Genes under Different Pst DC3000-Related Treatments during PTI and ETI

The interactions between pathogens and plants are complex, including PTI and ETI. Furthermore, to understand the expression patterns of 20 putative stresses-related *SlNAC* genes during PTI and ETI, four RNA-seq datasets were mined and analyzed, as shown in Figure 5. Figure 5a is based on the domain-wise effect of AvrPto on the tomato transcriptome 6 h post inoculation. AvrPto, as a kind of effector, can interfere with PTI and then activate the ETI of plants. AvrPto has two domains (CD loop and carboxyl-terminal domain, CTD). Once these domains were mutated, AvrPto cannot activate ETI. Figure 5a consists of two types of treatments: D29E (pCPP45::avrPto I96A), representing *Pst* DC3000, whose CD circle was mutated and D29E (pCPP45::avrPto 2xA), representing *Pst* DC3000, whose CTD was mutated. Figure 5a shows that the expression levels of *SSlNAC1*, *SSlNAC3*, *SSlNAC4*, *SSlNAC6*, *SSlNAC7*, *SSlNAC8*, *SSlNAC10*, *SSlNAC12*, *SSlNAC17*, and *SSlNAC19* were higher in mutated AvrPto, which means that these genes were up-regulated during PTI. Figure 5b was based on transcriptome sequencing of tomato leaves treated with different *Pst* DC3000 mutant strains. It can be seen that the expression levels of *SSlNAC1*, *SSlNAC6*, *SSlNAC8*, and *SSlNAC10* were significantly higher in DC3000ΔhopQ1-1ΔavrPtoΔavrPtoB than in DC3000ΔhopQ1-1ΔavrPtoΔavrPtoBΔfliC, which means that these genes were up-regulated during PTI. Figure 5c is based on transcriptome sequencing of leaves of resistant (RG-PtoR) and susceptible (RG-prf3 and RG-prf19) tomato plants treated with *Pst* DC3000, and leaf samples were collected at 4 and 6 h after inoculation. PtoR represents tomatoes with the normal Pto/Prf signaling pathway, and once these tomatoes were infected by pathogens, then ETI could be induced. Prf3 and Prf19 represent tomatoes whose Prf was mutated, and ETI could not be induced after being infected by pathogens. Figure 5c illustrates that the expression levels of *SSlNAC3*, *SSlNAC4*, *SSlNAC7*, *SSlNAC8*, *SSlNAC11*, *SSlNAC12*, *SSlNAC14*, *SSlNAC1*7, *SSlNAC19,* and *SSlNAC20* were obviously increased in PtoR compared to Prf3 or Prf19, which means that these genes were up-regulated during ETI. The expression levels of *SSlNAC1*, *SSlNAC6*, and *SSlNAC10* were decreased in PtoR compared with Prf3 or Prf19, which means that these genes were down-regulated during ETI. Figure 5d is based on the transcriptome sequencing of tomato leaves treated with flgII-28 and DC3000ΔavrPtoΔavrPtoB. FlgII-28 was encoded by *flic* and can be recognized by receptor FLS2, which could induce PTI. However, PTI could be overcome by the T3SS (type III secretion system) of pathogens, including AvrPto and AvrPtoB. Therefore, if AvrPto and AvrPtoB mutated, ETI could not be induced. Figure 5d shows that the expression levels of *SSlNAC1*, *SlNAC3*, *SlNAC4*, *SSlNAC6*, *SSlNAC8, SSlNAC10, SSlNAC17*, and *SSlNAC19* were obviously increased under FlgII-28 treatment and DC3000ΔavrPtoΔavrPtoB treatment, which means these *SSlNAC* genes are up-regulated during PTI. All the information about the RNA-seq data used in this study is summarized in Appendix A, including the treatments, plant genotypes, and times post-treatment, etc.

To conclude, it can be seen that *SSlNAC3*, *SSlNAC4*, *SSlNAC7*, *SSlNAC8*, *SSlNAC12*, *SSlNAC17*, and *SSlNAC19* were up-regulated during both PTI and ETI under *Pst* DC3000 treatment, while *SSlNAC1*, *SSlNAC6*, and *SSlNAC10* were up-regulated only during PTI and *SSlNAC11*, *SSlNAC14*, and *SSlNAC20* were only up-regulated during ETI. In addition, *SSlNAC1*, *SSlNAC6*, and *SSlNAC10* were down-regulated during ETI (Appendix A, Figure 5 and Figure 6).

### 3.7. Expression Patterns of 20 SSlNAC Genes under Phytohormones Treatment

To explore the expression patterns of the 20 *SSlNAC* genes under phytohormone treatment, 4-week-old tomatoes were sprayed with IAA, ABA, SA, and MeJA. Then, RT-qPCR was carried out to detect the expression profiles of the 20 *SSlNAC* genes. Results showed that the expression levels of 19 *SSlNAC* genes (except for *SSlNAC19*) were significantly responsive to IAA treatment. *SSlNAC1* and *SSlNAC7* could be significantly induced at 3, 6, and 12 h under IAA treatment. The expression level of *SSlNAC10* was down-regulated at 6 h (Figure 7a). After treatment with ABA, *SSlNAC6* was induced at 3 h and 12 h, while it was suppressed at 6 h, thus indicating that it may play different roles in different stages of stress response. Expression levels of *SSlNAC10*, *SSlNAC11*, and *SSlNAC14* showed no significant difference compared with the control. The expression levels *SSlNAC8* and *SSlNAC19* were down-regulated (Figure 7b). It was further showed that the expression levels of 17 *SSlNAC* genes (except for *SSlNAC10*, *SSlNAC18*, and *SSlNAC19*) were significantly induced at different stages after treatment with MeJA, while 16 of them were up-regulated at 3 h, indicating that most of the *SSlNAC* genes were relatively early responsive genes (Figure 7c). Figure 7d shows that 18 *SSlNAC* genes (except for *SSlNAC5* and *SSlNAC19*) were remarkably induced after treatment with SA, while the expression levels of *SSlNAC10* and *SSlNAC13* were down-regulated. Among the 16 up-regulated *SSlNAC* genes, only *SSlNAC12* and *SSlNAC15* had the highest expression levels at 12 h after spraying SA, indicating these two genes were relatively late responsive genes.

## 4. Discussion

In recent years, the vegetable industry has been flourishing, with acreages of vegetables having markedly increased. Tomato, as one of the most consumed and economically important vegetables worldwide [35], is susceptible to various pathogens, especially to *Pst* DC3000. Once a tomato is infected by this biotrophic bacterial pathogen, tomato leaf spot disease could manifest, which finally will result in yield loss and reduced quality [49]. NAC TFs, as one of the largest TF families in plants, can be induced under biotic and abiotic stress, as well as phytohormones to regulate downstream defense genes. However, functional surveys of the *NAC* genes of tomato remain largely unstudied [66]. In this study, a comprehensive analysis of 101 *SlNAC* genes was carried out. Twenty *SlNAC* genes were identified as stress-related genes according to the result of the phylogenetic tree of the *NAC* genes. Furthermore, 18 of 20 *SSlNAC* genes were significantly induced in response to *Pst* DC3000 based on RT-qPCR analysis. In previous studies, three *NAC* genes, including *SlSRN1* (Solyc12g056790) [46], *SlNAC1* (Solyc04g009440) [51], and *JA2L* (Solyc07g063410) [37], were demonstrated to respond to *Pst* DC3000. *SlNAC1* and *JA2L* were also identified as stress-related genes in this study, namely, *SSlNAC4* and *SSlNAC12*. Besides, expression of most *SSlNAC* genes were induced by *Pst* DC3000 in 6–12 hpi (Figure 1), which indicates that these genes may be early pathogen-responsive genes, which is similar to the expression patterns of *SlSRN1* and *SlNAC1*.

A previous study proposed that many proteins changed during PTI or ETI, and some responses were triggered by both PTI and ETI [67]. Furthermore, expression patterns of the 20 *SSlNAC* genes during PTI or ETI were demonstrated. The results showed that 7 *SSlNAC* genes (*SSlNAC3*, *SSlNAC4*, *SSlNAC7*, *SSlNAC8*, *SSlNAC12*, *SSlNAC17*, and *SSlNAC19*) were significantly induced under *Pst* DC3000 treatment during both PTI and ETI, which is consistent with a previous report in which the expression of the majority of genes was induced during both PTI and ETI of the defense response [68] and a substantial overlap between the genes induced by flg22 (PTI) and those induced by effector recognition (ETI) was observed [69]. Three *SSlNAC* genes (*SSlNAC1*, *SSlNAC6*, and *SSlNAC10*) were only involved in defense response against *Pst* DC3000 during PTI, and these three genes were also showed to be down-regulated during ETI, which may indicate that these genes play different roles during the different phases of immunity when the plants were under *Pst* DC3000 stress.

Many of NAC transcription factors have been validated to be induced by exogenous phytohormones [37,46]. In this study, the expression profiles of 20 *SSlNAC* genes were obtained after spraying IAA, ABA, SA, and MeJA, which are well-known as defense-signaling hormones. The results showed that the expression levels of most of these *SSlNAC* genes were obviously increased. For instance, 17 *SSlNAC* genes (except for *SSlNAC10, SSlNAC11*, and *SSlNAC14*) were significantly induced after spraying ABA, while 10 of them were also induced under *Pst* DC3000 inoculation, including *SSlNAC1*, *SSlNAC4*, *SSlNAC6*, *SSlNAC9*, *SSlNAC12*, *SSlNAC13*, *SSlNAC16*, *SSlNAC17*, *SSlNAC18*, and *SSlNAC20*. Since ABA can regulate disease resistance via the ABA signaling pathway [34], we think the aforementioned 10 *SSlNAC* genes may regulate the interaction between tomato and *Pst* DC3000 through an ABA-dependent signaling network, although this speculation needs to be verified. For example, *SSlNAC4, SSlNAC12*, and *SSlNAC17* were significantly induced after ABA treatment, while they were also proven to be up-regulated during PTI and ETI, which indicated that they may positively regulate defense against *Pst* DC3000 through the ABA signaling pathway in both PTI and ETI. *SSlNAC20* was significantly induced at 3 and 12 h after ABA treatment, while it was up-regulated during only ETI, which indicate that *SSlNAC20* may positively regulate defense against *Pst* DC3000 through the ABA signaling pathway in only ETI. In addition, expression of most genes of the 20 *SSlNAC* genes were induced by IAA, SA, and MeJA, a result consistent with previous studies, and could be explained in that hormone-responsive elements exist in the promoters of *NAC* genes. These findings implied that *SSlNAC* genes participated in the *Pst* DC3000 stress response by multiple regulatory mechanisms.

## 5. Conclusions

To conclude, NAC transcription factors play important roles in stress responses and have been paid much attention recently, although the expression levels, function surveys, and regulatory mechanisms of most *SlNAC* genes resisting *Pst* DC3000 in tomato are still largely unstudied. In this study, a complete overview of the NAC gene family in tomato is presented. Among the 101 members of the *SlNAC* gene family, 20 *SSlNAC* genes were identified. Expression profile analysis revealed that 18 *SSlNAC* genes were significantly induced in defense response to *Pst* DC3000. In addition, the RNA-seq data analysis showed that 13 *SSlNAC* genes responded to *Pst* DC3000 during the PTI or ETI. Among them, *SSlNAC3*, *SSlNAC4*, *SSlNAC7*, *SSlNAC8*, *SSlNAC12*, *SSlNAC17*, and *SSlNAC19* were up-regulated in response to *Pst* DC3000 during both PTI and ETI, which suggests that these genes are involved in both the PTI and ETI pathways during the interaction between tomato and *Pst* DC3000. Furthermore, most of the *SSlNAC* genes can be induced under IAA, SA, MeJA, and ABA treatment, which indicate that tomato NAC TFs may resist *Pst* DC3000 through complex phytohormone signaling pathways. Taken together, this study revealed the differential expression patterns of 20 *SSlNACs* under *Pst* DC3000 stress and selected candidate genes for further functional analysis. Furthermore, this study provides a basis for analysis of the regulatory mechanism under biotic stress.

## Figures and Tables

**Figure 1 genes-13-02015-f001:**
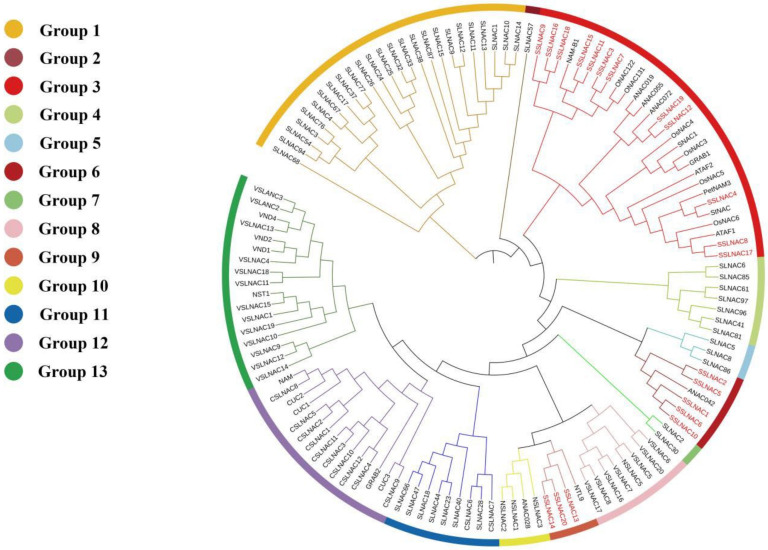
The phylogenetic tree of the 101 *NAC* genes in tomato and the *NAC* genes related to biotic/abiotic stresses in rice and *Arabidopsis.* The neighbor-joining (NJ) phylogenetic tree was constructed using MEGA software version 5.05 with 1000 bootstrap replicates. They are divided into 13 groups, labeled with different colors, and the 20 putative stress *SlNAC* genes (*SSlNAC1*–*20*) are indicated in red font.

**Figure 2 genes-13-02015-f002:**
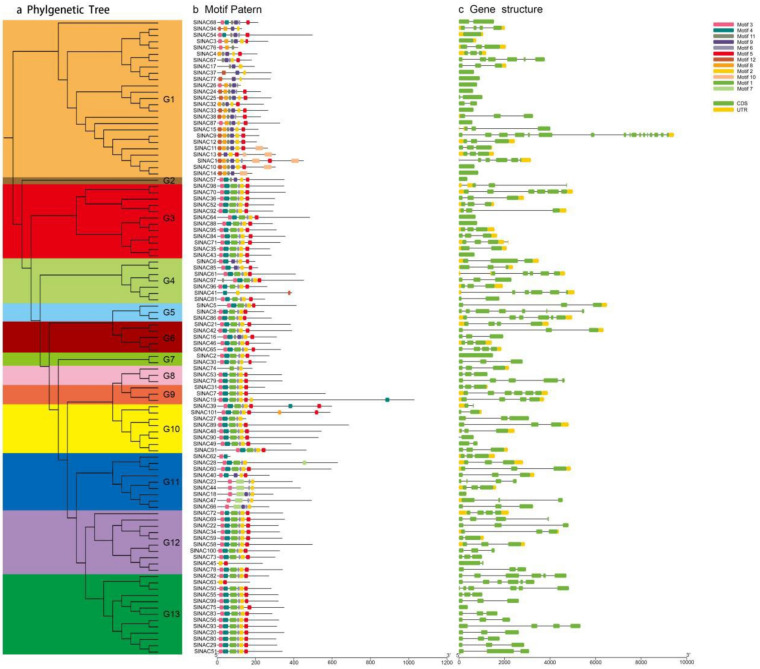
Phylogenetic tree, conserved motifs, and gene structures of the *SlNAC* genes. (**a**) Phylogenetic tree constructed using MEGA software version 5.05 with 1000 bootstrap replicates based on 101 SlNAC protein full-length sequences (G1–G13 indicate 13 clades). (**b**) The motif compositions of the SlNACs were identified using the Multiple Expectation Maximization for Motif Elicitation (MEME) program. A total of 12 motifs were predicted and the different motifs are indicated in different colors. (**c**) The gene structures of the *SlNACs* were visualized using TBtools software version 1.098769. The green boxes represent CDs, the yellow boxes represent UTR, and the black lines represent introns.

**Figure 3 genes-13-02015-f003:**
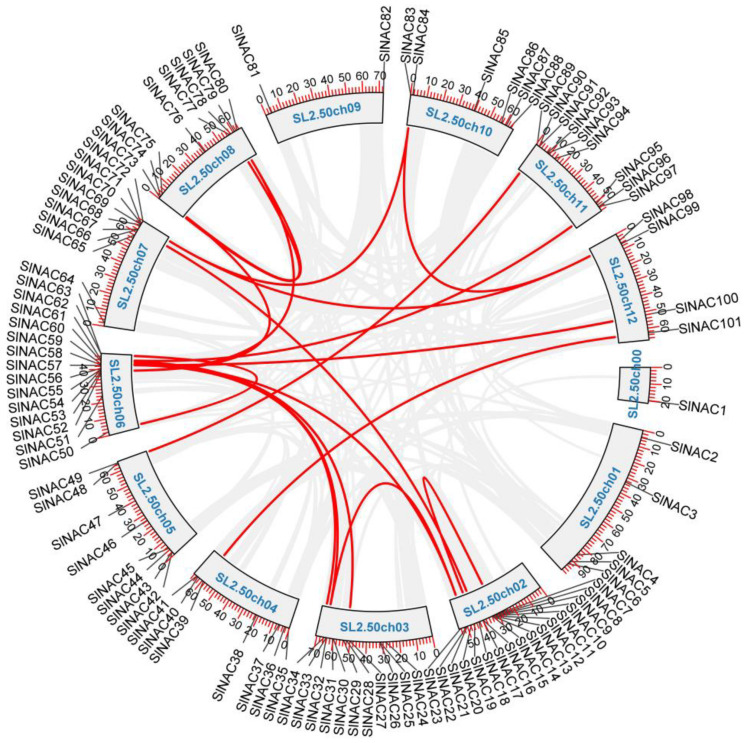
Chromosomal distribution and collinear relationships of the *SlNAC* genes. The duplicated gene pairs are joined by red lines.

**Figure 4 genes-13-02015-f004:**
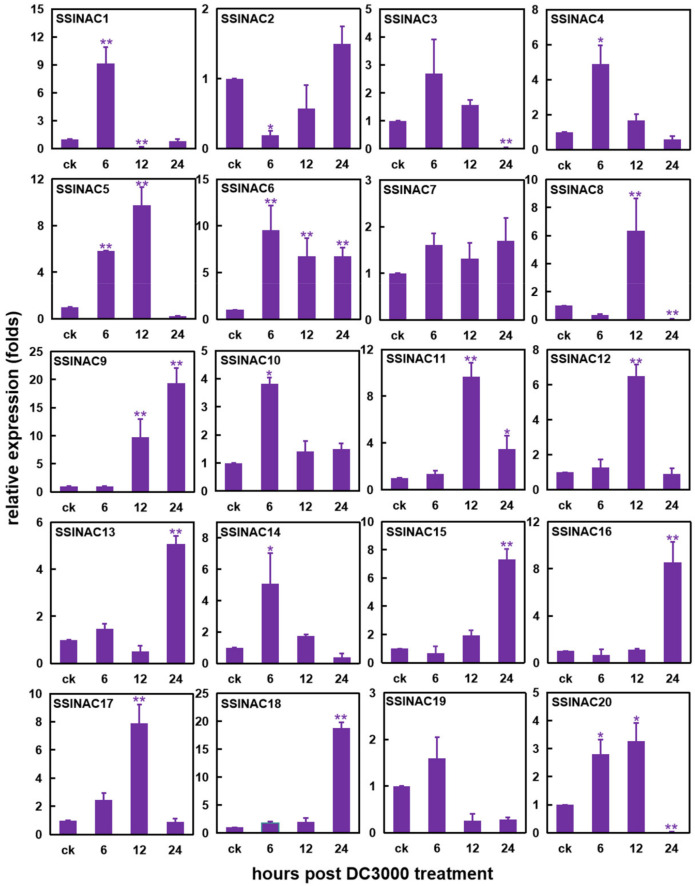
Expression levels of *SSlNAC1–20* at 6, 12, and 24 h post *Pseudomonas syringae* pv. *tomato* DC3000 (*Pst* DC3000) inoculation. The 2^−ΔΔCt^ method was used to calculate the expression fold for each *SSlNAC* gene. Data are the mean ± standard errors (SD) from three independent biological replicates. Error bars indicate standard error. An asterisk (*) represents a significant difference (* *p* < 0.05, ** *p* < 0.01).

**Figure 5 genes-13-02015-f005:**
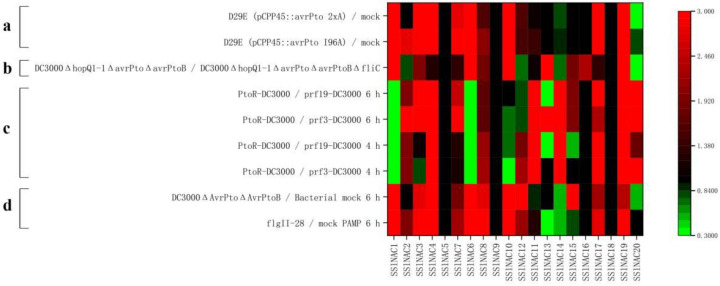
The heat map represents the expression of 20 *SSlNAC* genes in response to *Pst* DC3000 with different treatments. (**a**,**c**,**d**) Expression profiles of 20 *SSlNAC* genes under *Pst* DC3000 mutant strains. (**b**) Expression profiles of 20 *SSlNAC* genes in resistant and susceptible tomato under *Pst* DC3000 treatment.

**Figure 6 genes-13-02015-f006:**
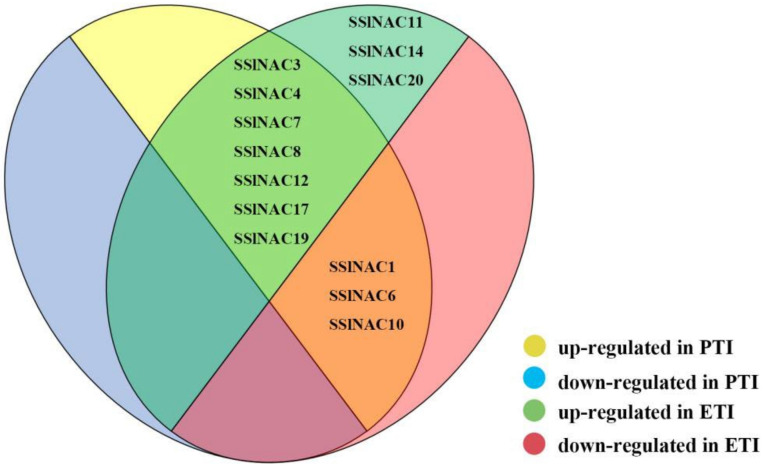
Venn diagram showing the expression patterns of the 20 *SSlNAC* genes during PTI and ETI under *Pst* DC3000 stress.

**Figure 7 genes-13-02015-f007:**
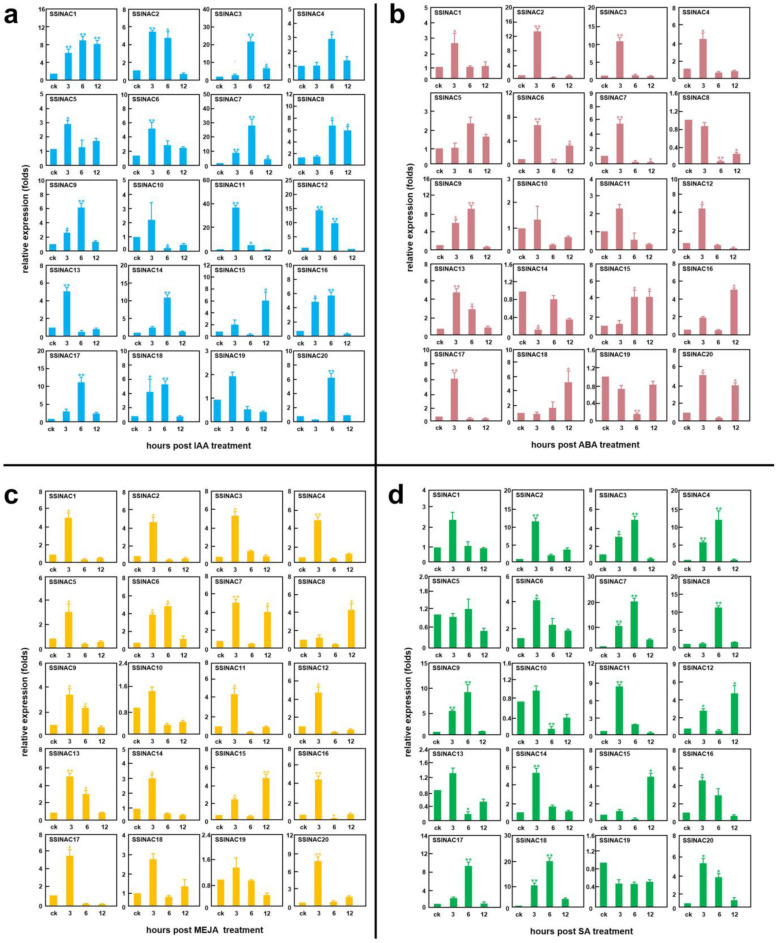
Expression levels of *SSlNAC1–20* under different phytohormone treatments at different times (3, 6, and 12 h): (**a**) IAA (indole-3-acetic acid) treatment; (**b**) ABA (abscisic acid) treatment; (**c**) MEJA (methyl jasmonic acid) treatment; (**d**) SA (salicylic acid) treatment. The 2^−ΔΔCt^ method was used to calculate the expression fold for each *SSlNAC* gene. Data are the mean ± standard errors (SD) from three independent biological replicates. Error bars indicate the standard error. An asterisk (*) represents a significant difference (* *p* < 0.05, ** *p* < 0.01).

## Data Availability

Not applicable.

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
