# Peer review of "Comprehensive Analysis of NAC Genes Reveals Differential Expression Patterns in Response to Pst DC3000 and Their Overlapping Expression Pattern during PTI and ETI in Tomato"

_genes, 2022, doi:10.3390/genes13112015_

Round 1
Reviewer 1 Report
In the submitted manuscript by Xu et al., entitled “Comprehensive Analysis of NAC Genes Reveals Differential Expression Patterns in Response to Pst DC3000 and their Overlapping Expression Pattern during PTI and ETI in Tomato” the authors studied the expression pattern of 20 SSINAC genes in response to Pst DC3000 based on previously available data. The authors identified stress-responsive SSINACs and determined the expression pattern of these genes during PTI and ETI. Moreover, using qRT PCR, they elucidated the expression pattern of these SSINAC genes under phytohormone treatment.
Overall, the experiments included in this manuscript are informative, and generally, the interpretations seem reasonable to a certain extent. However, as I detail below, there are several issues with the figures that should be addressed.
1. Fig. 1 and 2: Please make them full-page figures with better resolution. Figures have very low resolution, you can’t even properly see the color pattern. In fig. 2 even the color coding at the upper right-hand side is extremely hazy and not readable.
2. Fig. 3: The chromosome number (I guess) written with yellow color is not at all visible and has poor resolution.
3. Fig. 4 & 7: In addition to the text please provide the information about the graph and experiment in the figure legend e.g. internal control, normalization, replicates, error bars, etc. Fig. 7 is too small, please make it bigger.
4. Fig. 5: Again very poor resolution, it’s difficult to understand and interpret anything written or shown in the figure.
Author Response
Response to Reviewer 1 Comments
Point 1: Fig. 1 and 2: Please make them full-page figures with better resolution. Figures have very low resolution, you can’t even properly see the color pattern. In fig. 2 even the color coding at the upper right-hand side is extremely hazy and not readable.
Response 1: We have revised our manuscript as this comment. Resolutions of Fig. 1 and 2 have been improved, and detailed information can be seen by amplifying these figures.
Point 2: Fig. 3: The chromosome number (I guess) written with yellow color is not at all visible and has poor resolution.
Response 2: We have revised our manuscript as this comment.
Point 3: Fig. 4 & 7: In addition to the text please provide the information about the graph and experiment in the figure legend e.g. internal control, normalization, replicates, error bars, etc. Fig. 7 is too small, please make it bigger.
Response 3: We have revised our manuscript as this comment.
Point 4: Fig. 5: Again very poor resolution, it’s difficult to understand and interpret anything written or shown in the figure.
Response 4: We have revised our manuscript as this comment. Resolution of Fig. 5 has been improved, and detailed information can be seen by amplifying this figure.

Reviewer 2 Report
Authors of this manuscript titled “Comprehensive Analysis of NAC Genes Reveals Differential Expression Patterns in Response to Pst DC3000 and their Overlapping Expression Pattern during PTI and ETI in Tomato” characterise a selection of predicted stress-associated NAC transcription factors. The writing is generally good but several grammatical errors throughout the manuscript exist and the manuscript should be submitted to a English language editing service prior to resubmission.
This study is suitable for this journal and is of considerable interest. Authors should carefully consider the following comments and changes to make the manuscript more accurate and acceptable for publication:
Major changes:
1. It is unclear whether the Suhong 3 cultivar used in this study is susceptible or resistant to Pst DC3000. Please make this clear and show disease symptoms (or lack thereof) of infected plants to demonstrate susceptibility or resistance.
2. The so called “microarray data” used in this study is largely RNA sequencing studies it appears. These need to be referenced appropriately and more extensively/accurately explained regarding treatments, plant genotypes and times post-treatment that the RNA seq data represents for each study. (See comment below about PTI and ETI).
3. The figure legends throughout this manuscript are very poorly written with insufficient details to interpret the figure.
4. The interpretation of PTI and ETI-specific activation of NAC expression (lines 273-327, including figures) is quite problematic, for several reasons. This whole section needs to be rewritten. Firstly, the supplied Figure 5 is not legible. No interpretation using this figure can currently be made. Secondly, the relevant studies that produced the data in Figure 5 have not been referenced and therefore the treatments cannot be validated AND the figure legend does not carry any of the essential information. Third, no P-values are reported for these analyses so their statistical significance cannot be determined. Fourth, the strains that are expected to trigger PTI (e.g. Pst DC3000 ΔavrPtoΔavrPtoB and Pst DC3000 D28E) are confused with strains that should trigger little to no PTI and instead are suggested to trigger ETI for unknown reasons (e.g. Pst DC3000 ΔavrPtoΔavrPtoBΔfliC and Pst DC3000 ΔhopQ1-1ΔfliC). Notably, the Pst DC3000 ΔhopQ1-1 background is only really interesting in N. benthamiana for not triggering ETI. Strains that trigger ETI (e.g. Pst DC3000 wildtype in introgressed Pto-carrying plants, RG-PtoR) are also not appropriately compared with strains that also should trigger ETI (e.g. Pst DC3000 D28E carrying avrPto). A lot of unnecessary mutants’ data are included as well, for example the Pst D28E avrPto I96A, Pst D28E 2xA, and Pst D28E I96A/2xA double mutant. These are too convoluted to add value to the analysis. Sentences in lines 289-291 and lines 296-305 are the only truly factual statements, but again the data for this is not validated. The sentence in lines 317 to 320 is too general. The NACs should be broken down into four groups: Upregulated in PTI, downregulated in PTI, upregulated in ETI, and downregulated in ETI. There may be some overlap of NACs across these four groups.
A table of the strains and plant genotype (pto or Pto-R) combinations absolutely needs to be made and presented to show what combinations are leading to predicted PTI and/or ETI responses. This table should also summarize NACs that are upregulated or downregulated by that PTI/ETI response. The sentence in lines 317 to 320 is too general. The NACs should be broken down into four groups: Upregulated in PTI, downregulated in PTI, upregulated in ETI, and downregulated in ETI. There may be some overlap of NACs across these four groups. Figure 6 should also be modified to reflect the up/down status of these genes’ expression.
5. Figure 7 is of very poor quality and cannot be interpreted. I cannot see statistical significance reporting and the y-axis scale is unreadable.
Furthermore, additional analysis of comparing the hormone responsive NACs to the PTI/ETI inducible NACs could suggest relationships between Pst DC3000 defence and phytohormones. A comparison of key NACs that mediate this relationship would be very valuable to the work.
Minor changes:
Line 16-17: Should just say Pseudomonas syringae pv. tomato (the DC3000 strain is a mutant of a wild isolate)
Line 80-82: How is JA2L relevant to NACs?
Line 94-95: Pseudomonas syringae pv. tomato DC3000 (Pst DC3000) description should be earlier in the manuscript
Line 97-98: Sentence beginning “To dissect…” is incomprehensible. Grammar correction needed.
Line 132: “Collinerity” > “Collinearity”
Line 331: “files” > “profiles”
Line 332: “respond” > “responsive”
Line 367: Should just say P. syringae pv. tomato and not DC3000
Line 368: “biotroph bacteria disease” > “biotrophic bacterial pathogen”
Author Response
Response to Reviewer 2 Comments
The writing is generally good but several grammatical errors throughout the manuscript exist and the manuscript should be submitted to a English language editing service prior to resubmission.
Response: Because of the deadline, we have submitted our manuscript to Dr. Xue, who have studied abroad for many years, for English language editing service.
Major changes:
Point 1: It is unclear whether the Suhong 3 cultivar used in this study is susceptible or resistant to Pst DC3000. Please make this clear and show disease symptoms (or lack thereof) of infected plants to demonstrate susceptibility or resistance.
Response 1: Suhong 2003 was bought from Jiangsu Academy of Agricultural Science. It is known to resist Tomato Masaic Virus (ToMV), Tomato Leaf Mould and Tomato Fusarium wilt; however, its resistance to Pst DC3000 is unknown.
Point 2: The so called “microarray data” used in this study is largely RNA sequencing studies it appears. These need to be referenced appropriately and more extensively/accurately explained regarding treatments, plant genotypes and times post-treatment that the RNA seq data represents for each study. (See comment below about PTI and ETI).
Response 2: The relevant studies that produced the data in Figure 5 have been referenced in our revised manuscript. Besides, all these information about microarray data used in this study has been summarized in Table S3, including treatments, plant genotypes and times post-treatment, etc.
Point 3: The figure legends throughout this manuscript are very poorly written with insufficient details to interpret the figure.
Response 3: We have revised our manuscript as this comment.
Point 4: The interpretation of PTI and ETI-specific activation of NAC expression (lines 273-327, including figures) is quite problematic, for several reasons. This whole section needs to be rewritten.
Response: We have rewritten this section.
Firstly, the supplied Figure 5 is not legible. No interpretation using this figure can currently be made.
Response: We have modified Figure 5.
Secondly, the relevant studies that produced the data in Figure 5 have not been referenced and therefore the treatments cannot be validated AND the figure legend does not carry any of the essential information.
Response: The relevant studies that produced the data in Figure 5 have been referenced in our revised manuscript.
Third, no P-values are reported for these analyses so their statistical significance cannot be determined.
Response: The p-values for these analyses have been used to draw conclusions (** p < 0.01, * p < 0.05).
Fourth, the strains that are expected to trigger PTI (e.g. Pst DC3000 ΔavrPtoΔavrPtoB and Pst DC3000 D28E) are confused with strains that should trigger little to no PTI and instead are suggested to trigger ETI for unknown reasons (e.g. Pst DC3000 ΔavrPtoΔavrPtoBΔfliC and Pst DC3000 ΔhopQ1-1ΔfliC). Notably, the Pst DC3000 ΔhopQ1-1 background is only really interesting in N. benthamiana for not triggering ETI. Strains that trigger ETI (e.g. Pst DC3000 wildtype in introgressed Pto-carrying plants, RG-PtoR) are also not appropriately compared with strains that also should trigger ETI (e.g. Pst DC3000 D28E carrying avrPto). A lot of unnecessary mutants’ data are included as well, for example the Pst D28E avrPto I96A, Pst D28E 2xA, and Pst D28E I96A/2xA double mutant. These are too convoluted to add value to the analysis. Sentences in lines 289-291 and lines 296-305 are the only truly factual statements, but again the data for this is not validated. The sentence in lines 317 to 320 is too general. The NACs should be broken down into four groups: Upregulated in PTI, downregulated in PTI, upregulated in ETI, and downregulated in ETI. There may be some overlap of NACs across these four groups.
Response: We have re-screened the data and unnecessary mutants’ data has been deleted, especially for the double mutants’ data. Then, some confused parts have also been corrected. Finally, we have broken these 20 SSlNAC genes into four groups as this comment.
A table of the strains and plant genotype (pto or Pto-R) combinations absolutely needs to be made and presented to show what combinations are leading to predicted PTI and/or ETI responses. This table should also summarize NACs that are upregulated or downregulated by that PTI/ETI response. The sentence in lines 317 to 320 is too general. The NACs should be broken down into four groups: Upregulated in PTI, downregulated in PTI, upregulated in ETI, and downregulated in ETI. There may be some overlap of NACs across these four groups. Figure 6 should also be modified to reflect the up/down status of these genes’ expression.
Response: A table (Table S3) has been made and all of the information mentioned in this comment has been included.
Point 5: Figure 7 is of very poor quality and cannot be interpreted. I cannot see statistical significance reporting and the y-axis scale is unreadable.
Furthermore, additional analysis of comparing the hormone responsive NACs to the PTI/ETI inducible NACs could suggest relationships between Pst DC3000 defence and phytohormones. A comparison of key NACs that mediate this relationship would be very valuable to the work.
Response 5: The quality of Figure 7 has been improved. Furthermore, we have added additional analysis of comparing the hormone responsive NACs to the PTI/ETI inducible NACs in the ‘Discussion’ part.
Minor changes:
Point 1: Line 16-17: Should just say Pseudomonas syringae pv. tomato (the DC3000 strain is a mutant of a wild isolate)
Response 1: No, Pseudomonas syringae pv. tomato DC3000 (Pst DC3000) is a variety of Pseudomonas syringae.
Point 2: Line 80-82: How is JA2L relevant to NACs?
Response 2: JA2L is a member of NAC transcription factors family. The location of JA2L (Solyc07g063410) is also corresponding to SSlNAC12.
Point 3: Line 94-95: Pseudomonas syringae pv. tomato DC3000 (Pst DC3000) description should be earlier in the manuscript.
Response 3: We have mentioned Pseudomonas syringae pv. tomato DC3000 (Pst DC3000) description in the ‘Abstract’, and the description in line 94-95 is the first time in the ‘Introduction’.
Point 4: Line 97-98: Sentence beginning “To dissect…” is incomprehensible. Grammar correction needed.
Response 4: We have revised this sentence as this comment.
Point 5: Line 132: “Collinerity” > “Collinearity”
Response 5: We have revised it as this comment.
Point 6: Line 331: “files” > “profiles”
Response 6: We have revised it as this comment.
Point 7: Line 332: “respond” > “responsive”
Response 7: We have revised it as this comment.
Point 8: Line 367: Should just say P. syringae pv. tomato and not DC3000
Response 8: No, Pseudomonas syringae pv. tomato DC3000 (Pst DC3000) is a variety of Pseudomonas syringae.
Point 9: Line 368: “biotroph bacteria disease” > “biotrophic bacterial pathogen”
Response 9: We have revised it as this comment.

Round 2
Reviewer 1 Report
I have no further comments on the manuscript.
Author Response
Reviwer 1 has no further comments on the manuscript.
Reviewer 2 Report
Please address all comments in attached document
Major changes:
Point 1: It is unclear whether the Suhong 3 cultivar used in this study is susceptible or resistant to Pst DC3000. Please make this clear and show disease symptoms (or lack thereof) of infected plants to demonstrate susceptibility or resistance.
Authors’ Response 1: Suhong 2003 was bought from Jiangsu Academy of Agricultural Science. It is known to resist Tomato Masaic Virus (ToMV), Tomato Leaf Mould and Tomato Fusarium wilt; however, its resistance to Pst DC3000 is unknown.
- This is exactly the point. If the outcome of the plant’s interaction with this pathogen is not known how can the author’s use this pathogen and then claim the response as being PTI, ETI, or ETS? Without testing the disease outcome, the data from these experiments cannot be acceptable.
Point 2: The so called “microarray data” used in this study is largely RNA sequencing studies it appears. These need to be referenced appropriately and more extensively/accurately explained regarding treatments, plant genotypes and times post-treatment that the RNA seq data represents for each study. (See comment below about PTI and ETI).
Authors’ Response 2: The relevant studies that produced the data in Figure 5 have been referenced in our revised manuscript. Besides, all these information about microarray data used in this study has been summarized in Table S3, including treatments, plant genotypes and times post-treatment, etc.
- Microarray data is not the same as RNAseq data. The technologies are different and therefore they need to be explained correctly. Just using “microarray data” to describe RNAseq data is inaccurate and unacceptable.
- Some of the data summarized in Table S3 is incorrect or convoluted. This suggests that the authors have not appropriately examined the strain-plant interactions. Therefore, their conclusions are flawed and need to be reviewed. Specifically:
o DC3000ΔhrcQ-UΔfliC /Bacterial mock 6 h = ETI
(This is incorrect. With the hrcQ-U locus deleted this strain should NOT trigger ETI. At best this is PTI but deletion of fliC renders this inconclusive. I would remove this dataset entirely as it is too complicated to conclude from.)
o D28E (pCPP45::avrPto I96A or other modifications) / mock = PTI
(Firstly, this data is referenced incorrectly as Yeam et al. It needs to be Worley et al., 2016 BMC Genomics and the strain used is actually D29E. While this conclusion is likely correct, a much better comparison would be ΔavrPtoΔavrPtoBΔfliC/ ΔavrPtoΔavrPtoB – this is in the D011 data set removed from previous analysis. Why include the AvrPto mutant comparisons when it is unnecessarily complicated? I strongly suggest removing most D013 avrPto data, perhaps keeping a single mutant. Then return this single comparison ΔavrPtoΔavrPtoBΔfliC/ ΔavrPtoΔavrPtoB from D011 data as PTI.)
Minor changes:
Point 1: Line 16-17: Should just say Pseudomonas syringae pv. tomato (the DC3000 strain is a mutant of a wild isolate)
Authors’ Response 1: No, Pseudomonas syringae pv. tomato DC3000 (Pst DC3000) is a variety of Pseudomonas syringae.
- No, this is not correct. DC3000 is a rifampicin mutant of a wild isolate. Therefore, you cannot say that DC3000 is causing disease. See: Cuppels, D. A. 1986. Generation and characterization of Tn5 insertion mutations in Pseudomonas syringae pv. tomato. Appl. Environ. Microbiol. 51:323-327
Point 2: Line 80-82: How is JA2L relevant to NACs?
Authors’ Response 2: JA2L is a member of NAC transcription factors family. The location of JA2L (Solyc07g063410) is also corresponding to SSlNAC12.
- Where this is mentioned in the text in Line 84 it is not clear that JA2L is a NAC. This is mentioned later in the manuscript (Line 102). Clearly indicate in the text that JA2L is a NAC.
Point 3: Line 94-95: Pseudomonas syringae pv. tomato DC3000 (Pst DC3000) description should be earlier in the manuscript.
Authors’ Response 3: We have mentioned Pseudomonas syringae pv. tomato DC3000 (Pst DC3000) description in the ‘Abstract’, and the description in line 94-95 is the first time in the ‘Introduction’.
- Where this is mentioned in the text in Line 84 it is abbreviated as Pto. Then it is later expanded in Line97-98. This a simple mistake to fix.
Point 8: Line 367: Should just say P. syringae pv. tomato and not DC3000
Authors’ Response 8: No, Pseudomonas syringae pv. tomato DC3000 (Pst DC3000) is a variety of Pseudomonas syringae.
- No, this is not correct. DC3000 is a rifampicin mutant of a wild isolate. Therefore, you cannot say that DC3000 is causing disease. See: Cuppels, D. A. 1986. Generation and characterization of Tn5 insertion mutations in Pseudomonas syringae pv. tomato. Appl. Environ. Microbiol. 51:323-327

Author Response
Response to Reviewer 2 Comments
Major changes:
Point 1: It is unclear whether the Suhong 3 cultivar used in this study is susceptible or resistant to Pst DC3000. Please make this clear and show disease symptoms (or lack thereof) of infected plants to demonstrate susceptibility or resistance.
Authors’ Response 1: Suhong 2003 was bought from Jiangsu Academy of Agricultural Science. It is known to resist Tomato Masaic Virus (ToMV), Tomato Leaf Mould and Tomato Fusarium wilt; however, its resistance to Pst DC3000 is unknown.
This is exactly the point. If the outcome of the plant’s interaction with this pathogen is not known how can the author’s use this pathogen and then claim the response as being PTI, ETI, or ETS? Without testing the disease outcome, the data from these experiments cannot be acceptable.
Response 1:
(1) The reason why we chose Suhong 2003 is that it has been used in several published studies about the interaction between tomato and pst DC3000. We also added this description in our revised manuscript, as follows: “According the previous studies [46,55,56], tomato (Solanum lycopersicum) cv. Suhong 2003 was used in this study. Tomato plants were grown in a growth chamber (24℃ 14h light/20℃ 10h dark). For analysis of gene expression in response to Pst DC3000, four-week-old tomato seedlings were infiltrated with Pst DC3000 suspensions according to previously reported procedure [46,55,56].”
(2) The reason why we did not test the disease outcome is that we actually did not come to conclusions about the response as being PTI or ETI by our own experiment, and we came to such conclusions by using the RNA-seq data from the public database.
(3) When we explored the expression levels of SlNAC genes under pst DC3000 treatment, disease symptom of tomato infected by Pst DC3000 was observed at the 3rd day after inoculation in our naked eyes. Considering that many similar papers about analysis of genes expression did not show disease symptom of infected plants [46, 55-59], we did not take photos either.
(4) There are some pictures in published papers about functional study of genes showing the disease symptom of tomato (Suhong 2003) infected by Pst DC3000 [46,55,56]. Here we showed a picture from Li et al., Front. Plant Sci.2015, 6, 717, Figure 3A [55].
- Liu,B.; Ouyang, Z.G.; Zhang, Y.F.; Li, X.H.; Hong, Y.B.; Huang, L.; Liu, S.X.; Zhang, H.J.; Li, D.Y.; Song, F.M. Tomato NAC transcription factor SlSRN1 positively regulates defense response against biotic stress but negatively regulates abiotic stress response. PLoS ONE 2014, 9, e102067.
- Li, X.; Huang, L.; Hong, Y.; Zhang, Y.; Liu, S.; Li, D.; Zhang, H.; Song, F. Co-silencing of tomato S-adenosylhomocysteine hydrolase genes confers increased immunity against Pseudomonas syringae tomato DC3000 and enhanced tolerance to drought stress. Front. Plant Sci. 2015, 6, 717.
- Li, X.; Huang, L.; Zhang, Y.; Ouyang, Z.; Hong, Y.; Zhang, H.; Li, D.; Song, F. Tomato SR/CAMTA transcription factors SlSR1 and SlSR3L negatively regulate disease resistance response and SlSR1L positively modulates drought stress tolerance.BMC Plant Biol. 2014, 14, 286.
Point 2: The so called “microarray data” used in this study is largely RNA sequencing studies it appears. These need to be referenced appropriately and more extensively/accurately explained regarding treatments, plant genotypes and times post-treatment that the RNA seq data represents for each study. (See comment below about PTI and ETI).
Authors’ Response 2: The relevant studies that produced the data in Figure 5 have been referenced in our revised manuscript. Besides, all these information about microarray data used in this study has been summarized in Table S3, including treatments, plant genotypes and times post-treatment, etc.
-Microarray data is not the same as RNAseq data. The technologies are different and therefore they need to be explained correctly. Just using “microarray data” to describe RNAseq data is inaccurate and unacceptable.
-Some of the data summarized in Table S3 is incorrect or convoluted. This suggests that the authors have not appropriately examined the strain-plant interactions. Therefore, their conclusions are flawed and need to be reviewed. Specifically:
DC3000ΔhrcQ-UΔfliC /Bacterial mock 6 h = ETI
(This is incorrect. With the hrcQ-U locus deleted this strain should NOT trigger ETI. At best this is PTI but deletion of fliC renders this inconclusive. I would remove this dataset entirely as it is too complicated to conclude from.)
D28E (pCPP45::avrPto I96A or other modifications) / mock = PTI
(Firstly, this data is referenced incorrectly as Yeam et al. It needs to be Worley et al., 2016 BMC Genomics and the strain used is actually D29E. While this conclusion is likely correct, a much better comparison would be ΔavrPtoΔavrPtoBΔfliC/ ΔavrPtoΔavrPtoB – this is in the D011 data set removed from previous analysis. Why include the AvrPto mutant comparisons when it is unnecessarily complicated? I strongly suggest removing most D013 avrPto data, perhaps keeping a single mutant. Then return this single comparison ΔavrPtoΔavrPtoBΔfliC/ ΔavrPtoΔavrPtoB from D011 data as PTI.)
Response 2: Thank for your suggestion, and we have revised our manuscript as this comment. “Microarray data” has been changed into “RNA seq data”. Besides, two data have been deleted, including DC3000ΔhrcQ-UΔfliC /Bacterial mock 6 h and D28E (pCPP45::avrPto I96A + 2xA) / mock. In addition, ΔavrPtoΔavrPtoBΔfliC/ ΔavrPtoΔavrPtoB from D011 data has been returned.
Minor changes:
Point 1: Line 16-17: Should just say Pseudomonas syringae pv. tomato (the DC3000 strain is a mutant of a wild isolate)
Authors’ Response 1: No, Pseudomonas syringae pv. tomato DC3000 (Pst DC3000) is a variety of Pseudomonas syringae.
-No, this is not correct. DC3000 is a rifampicin mutant of a wild isolate. Therefore, you cannot say that DC3000 is causing disease. See: Cuppels, D. A. 1986. Generation and characterization of Tn5 insertion mutations in Pseudomonas syringae pv. tomato. Appl. Environ. Microbiol. 51:323-327
Response 1: Thank for your suggestion. Pseudomonas syringae pv. tomato is a variety of Pseudomonas syringae, and Pseudomonas syringae pv. tomato (Pst) DC3000 is a type strains of Pseudomonas syringae pv. tomato, and it is usually used to study the molecular mechanism of interaction between plants and pathogens. In this study, Pst DC3000 is the pathogen we actually used to infect tomatoes.
Point 2: Line 80-82: How is JA2L relevant to NACs?
Authors’ Response 2: JA2L is a member of NAC transcription factors family. The location of JA2L (Solyc07g063410) is also corresponding to SSlNAC12.
-Where this is mentioned in the text in Line 84 it is not clear that JA2L is a NAC. This is mentioned later in the manuscript (Line 102). Clearly indicate in the text that JA2L is a NAC.
Response 2: Thank for your suggestion, and we have revised our manuscript as this comment.
Point 3: Line 94-95: Pseudomonas syringae pv. tomato DC3000 (Pst DC3000) description should be earlier in the manuscript.
Authors’ Response 3: We have mentioned Pseudomonas syringae pv. tomato DC3000 (Pst DC3000) description in the ‘Abstract’, and the description in line 94-95 is the first time in the ‘Introduction’.
-Where this is mentioned in the text in Line 84 it is abbreviated as Pto. Then it is later expanded in Line97-98. This a simple mistake to fix.
Response 3: Thank for your suggestion, and we have revised our manuscript as this comment.
Point 4: Line 367: Should just say P. syringae pv. tomato and not DC3000
Authors’ Response 8: No, Pseudomonas syringae pv. tomato DC3000 (Pst DC3000) is a variety of Pseudomonas syringae.
-No, this is not correct. DC3000 is a rifampicin mutant of a wild isolate. Therefore, you cannot say that DC3000 is causing disease. See: Cuppels, D. A. 1986. Generation and characterization of Tn5 insertion mutations in Pseudomonas syringae pv. tomato. Appl. Environ. Microbiol. 51:323-327
Response 4: Thank for your suggestion. Pseudomonas syringae pv. tomato is a variety of Pseudomonas syringae, and Pseudomonas syringae pv. tomato (Pst) DC3000 is a type strains of Pseudomonas syringae pv. tomato, and it is usually used to study the molecular mechanism of interaction between plants and pathogens. In this study, Pst DC3000 is the pathogen we actually used to infect tomatoes.